# The Antibacterial and Anti-Biofilm Activity of Metal Complexes Incorporating 3,6,9-Trioxaundecanedioate and 1,10-Phenanthroline Ligands in Clinical Isolates of *Pseudomonas aeruginosa* from Irish Cystic Fibrosis Patients

**DOI:** 10.3390/antibiotics9100674

**Published:** 2020-10-05

**Authors:** Megan O’Shaughnessy, Pauraic McCarron, Livia Viganor, Malachy McCann, Michael Devereux, Orla Howe

**Affiliations:** 1School of Biological and Health Sciences, Technological University Dublin-City Campus, 9, D09 V209 Dublin, Ireland; megan.oshaughnessy@tudublin.ie; 2Centre for Biomemtic and Therapeutic Research, FOCAS Research Institute, Technological University Dublin-City Campus, 9, D09 V209 Dublin, Ireland; pauraic.mccarron@tudublin.ie (P.M.); liviaviganor@gmail.com (L.V.); michael.devereux@tudublin.ie (M.D.); 3Chemistry Department, National University of Ireland-Maynooth, W23 F2K8 Kildare, Ireland; malachy.mccann@mu.ie

**Keywords:** *Pseudomonas aeruginosa*, biofilm, metal complexes, 3,6,9-trioxaundecanedioic acid, 1,10-phenanthroline

## Abstract

Chronic infections of *Pseudomonas aeruginosa* in the lungs of cystic fibrosis (CF) patients are problematic in Ireland where inherited CF is prevalent. The bacteria’s capacity to form a biofilm in its pathogenesis is highly virulent and leads to decreased susceptibility to most antibiotic treatments. Herein, we present the activity profiles of the Cu(II), Mn(II) and Ag(I) tdda-phen chelate complexes {[Cu(3,6,9-tdda)(phen)_2_]·3H_2_O·EtOH}_n_ (Cu-tdda-phen), {[Mn(3,6,9-tdda)(phen)_2_]·3H_2_O·EtOH}_n_ (Mn-tdda-phen) and [Ag_2_(3,6,9-tdda)(phen)_4_]·EtOH (Ag-tdda-phen) (tddaH_2_ = 3,6,9-trioxaundecanedioic acid; phen = 1,10-phenanthroline) towards clinical isolates of *P. aeruginosa* derived from Irish CF patients in comparison to two reference laboratory strains (ATCC 27853 and PAO1). The effects of the metal-tdda-phen complexes and gentamicin on planktonic growth, biofilm formation (pre-treatment) and mature biofilm (post-treatment) alone and in combination were investigated. The effects of the metal-tdda-phen complexes on the individual biofilm components; exopolysaccharide, extracellular DNA (eDNA), pyocyanin and pyoverdine are also presented. All three metal-tdda-phen complexes showed comparable and often superior activity to gentamicin in the CF strains, compared to their activities in the laboratory strains, with respect to both biofilm formation and established biofilms. Combination studies presented synergistic activity between all three complexes and gentamicin, particularly for the post-treatment of established mature biofilms, and was supported by the reduction of the individual biofilm components examined.

## 1. Introduction

*Pseudomonas aeruginosa* is a ubiquitous Gram-negative human pathogen that can colonise a diversity of sites with severe clinical consequences. Isolates of *P. aeruginosa* are found in nosocomial infections within immunocompromised patients, ranging from burn sepsis [1] to ventilator-associated pneumonia [2], to pulmonary infections of patients with lung afflictions such as cystic fibrosis (CF) and chronic obstructive pulmonary disease (COPD) [3]. CF is an inherited autosomal recessive disorder characterised by mutations in the cystic fibrosis transmembrane conductance regulator (CFTR) gene located on chromosome 7 that primarily affects the lungs. Of particular concern in Ireland is the ΔF508 mutation, of which 1 in 19 Irish people are asymptomatic carriers [4]. Individuals with this mutated gene suffer from a defective chloride channel protein on the epithelial cells of the lungs, causing an imbalance of chloride ions and fluid, and therefore the production of a thick mucus and impaired mucociliary clearance of inhaled microbes, resulting in recurrent chronic respiratory infections and inflammation. *P. aeruginosa* is the leading cause of infection in patients with CF, and it is more prevalent in adults [5].

The prosperity of this microorganism is attributed to its biofilm forming capability and its vast repertoire of cell-associated and extracellular virulence factors controlled by transcriptional regulators, which enables its swift adaptation to environmental changes and host defences. Biofilms are structured multi-cellular sessile communities, organized as micro-colonies, encased within a self-produced extracellular matrix (ECM) that forms the scaffold for its three-dimensional architecture [6]. This porous and complex structure is composed of water and extracellular polymeric substances which possess bacterial secreted biopolymers, such as polysaccharides, extracellular DNA (eDNA), proteins, lipids, and metabolites such as pyocyanin and pyoverdine [7]. The formation of a biofilm is a well-regulated, multi-step, endless cycle, including (i) initial reversible attachment of bacterial cells with biotic or abiotic surfaces, followed by (ii) irreversible attachment within quick succession leading to (iii) the development of the immature biofilm architecture as the bacteria undergoes numerous physiological and phenotypic changes. The aggregation and accumulation of adherent bacteria lead to (iv) the formation of multiple cell layers to form a mature biofilm with water channels responsible for the distribution of nutrients and signalling molecules within the biofilm. Finally, due to extrinsic or intrinsic factors, the bacteria (v) convert back to a planktonic state, to allow dispersal of the cells and colonization in a new cycle of biofilm formation [8,9,10].

Conventional antibiotics are active against planktonic cells of *P. aeruginosa* that cause acute infection, but often fail to completely eradicate their biofilms, leading to persistent infections. It is well documented that pathogenic biofilms from patients demonstrate up to 1000-fold reduced susceptibility in comparison to their planktonic counterparts, due to the protective and altered nature of the biofilm [11]. Formation of these aggregated communities with their inherent resistance to antibiotics and host immune attack are at the root of many persistent and chronic bacterial infections. For patients with CF who are infected with *P. aeruginosa*, antibiotic treatment often relieves the symptoms, but does not necessarily cure the infection [12]. The acquisition of chronic *P. aeruginosa* infection is associated with worsened disease progression and increased mortality. Moreover, the effectiveness of the antibacterial agents to treat these pathogens is also compromised by the emergence of multidrug-resistant (MDR) strains, which highlights the urgent need for novel drugs that can eradicate the robust biofilms of *P. aeruginosa*, by acting either alone or in tandem with other clinical therapeutic interventions.

Over the past two decades, transition metal complexes have had a revival of interest as possible alternatives or adjuvants to the current arsenal of antimicrobial agents, and in particular due to the rapid emergence of resistant microorganisms occurring worldwide [13]. Novel inorganic complexes with 1,10-phenanthroline (phen) ligands have demonstrated promising therapeutic capabilities with diverse biological activity, including anticancer [14,15], antifungal [16,17], antibacterial [18,19] and antiviral [20] capabilities. In a recent review, we have reported on the research that points to the potential of metal-phenanthroline complexes as alternative therapeutics in this era of antibiotic resistance [21]. For example, Viganor et al. [18] investigated the effect of phen, 1,10-phenathroline-5,6-dione (phendione) and the metal complexes [Cu(phendione)_3_]^2+^ and [Ag(phendione)_2_]^+^ on planktonic- and biofilm-growing clinical isolates of *P. aeruginosa*. The complexes and the metal free phen and phendione ligands exhibited potency against susceptible and resistant planktonic cells in the following order: [Cu(phendione)_3_]^2+^ (MIC = 7.76 µM) > [Ag(phendione)_2_]^+^ (MIC = 14.05 µM) > phendione (MIC = 31.15 µM) > phen (MIC = 579.28 µM). The pre-treatment of the isolates with phen, phendione and the metal-phendione complexes at the 0.5 × MIC value inhibited biofilm formation, significantly reducing both biomass and viability. The complexes could also disrupt an established biofilm in a dose-dependent manner. Further investigation demonstrated that the metal complexes were effective inhibitors of the metalloenzyme Elastase B (lasB), suggesting this as a potential therapeutic target [22]. In a more recent study, [Cu(phendione)_3_]^2+^ and [Ag(phendione)_2_]^+^ were also shown to exhibit clinically relevant antimicrobial action against planktonic- and biofilm-growing cells of carbapenemase-producing *Acinetobacter baumannii* clinical strains derived from hospital patients in Brazil [23].

We have also recently evaluated the antimicrobial potential of a range of Cu(II), Mn(II) and Ag(I) dicarboxylate-phen chelate complexes against nine clinical isolates of three *Candida haemulonii* species, an emerging opportunistic pathogen resistant to most current clinical antifungal drugs [17]. In that study, the Ag(I) chelates were shown to be the most effective drugs against planktonic cells (overall geometric mean of the minimum inhibitory concentration (GM-MIC) ranged from 0.26 to 2.16 µM), followed by the Mn(II) chelates (0.87–10.71 µM) and the Cu(II) chelates (3.37–72 µM). Although the same general level of activity was not maintained when the strains were tested during a biofilm lifecycle of *Candida haemulonii*, the Mn(II) chelates with different dicarboxylate ligands produced an overall geometric mean of the biofilm minimum inhibitory concentration (GM-bMIC) in the range 3.5–5.3 µM. Furthermore, the in vivo potential of the chelates to clear *C. haemulonii* infections using the *Galleria mellonella* model revealed that the Mn(II)- and Ag(I)-phen chelates were able to conserve antifungal activity at concentrations that were reasonably non-toxic toward *G. mellonella* [24]. Most notable was {[Mn(3,6,9-tdda)(phen)_2_]·3H_2_O·EtOH}_n_ (Mn-tdda-phen) (tddaH_2_ = 3,6,9-trioxaundecanedioic acid), as it induced the lowest *G. mellonella* mortality rate while reducing the fungal burden on infected larvae, and it also affected the virulence of *C. haemulonii.* Additionally, the presence of the extremely hydrophilic 3,6,9-trioxaundecanedioate ligand in the metal-tdda-phen complexes considerably increased their water solubility when compared to that of the complexes containing alternative aliphatic or aromatic dicarboxylate ligands.

Herein, we report the results of a further investigation into the antimicrobial activity of metal-tdda-phen complexes. The complexes {[Cu(3,6,9-tdda)(phen)_2_]·3H_2_O·EtOH}_n_ (Cu-tdda-phen), {[Mn(3,6,9-tdda)(phen)_2_]·3H_2_O·EtOH}_n_ (Mn-tdda-phen) and [Ag_2_(3,6,9-tdda)(phen)_4_]·EtOH (Ag-tdda-phen) were tested on three clinical isolates of *P. aeruginosa* (CF1-CF3), derived from Irish hospital patients with CF, both alone and in combination with gentamicin, a well-known antibiotic. The results were compared to those derived for two established laboratory strains (ATCC 27853 and PAO1). The effects on planktonic growth, as well as the inhibition of biofilm formation and biofilm disarticulation, were assessed in terms of biofilm biomass and cell viability once treated. The anti-biofilm effects were further probed through an analysis of individual biofilm components, including polysaccharides, eDNA and the virulence factors pyocyanin and pyoverdine, that help establish and maintain the biofilm.

## 2. Results

### 2.1. The Inhibition of Planktonic Bacterial Growth by Metal-Tdda-Phen Complexes

Antimicrobial susceptibility testing of {[Cu(3,6,9-tdda)(phen)_2_]·3H_2_O·EtOH}_n_ (Cu-tdda-phen), {[Mn(3,6,9-tdda)(phen)_2_]·3H_2_O·EtOH}_n_ (Mn-tdda-phen) and [Ag_2_(3,6,9-tdda)(phen)_4_]·EtOH (Ag-tdda-phen), along with their phen-free precursors, [Cu(3,6,9-tdda)]·H_2_O (Cu-tdda), [Mn(3,6,9-tdda)(H_2_O)_2_]·2H_2_O (Mn-tdda) and [Ag_2_(3,6,9-tdda]·2H_2_O (Ag-tdda), and their component synthetic starting materials, 3,6,9-trioxaundecanedioic acid (tddaH_2_), phen, the simple salts MnCl_2_, CuCl_2_ and AgNO_3_, and the antibiotic gentamicin were evaluated through establishing their minimum inhibitory concentration (MIC), in accordance with the European Committee on Antimicrobial Susceptibility Testing (EUCAST) guidelines [25], and the results are summarised in Table 1. In accordance with the EUCAST breakpoints, control strains ATCC 27853 and PAO1 were susceptible to the aminoglycoside antibiotic gentamicin with MICs of 1 µg/mL (1.7 µM) and 2 µg/mL (3.5 µM), respectively, while all the clinical isolates were deemed resistant to the antibiotic. When all of the strains were incubated in the presence of the Cu(II), Mn(II) and Ag(I) simple salts (effectively the free metal ions) or in the presence of the metal-free phen ligand, no clinically relevant effects against either control strains (ATCC 27853 and PAO1) or clinical isolates (CF1-CF3) were observed. A similar inactivity profile was witnessed for the phen-free precursor complexes, Cu-tdda, Mn-tdda and Ag-tdda.

The metal-tdda-phen complexes were the most active against all of the *P. aeruginosa* strains compared to their non-phen precursors, simple metal salts and metal-free phen. The three metal-tdda-phen complexes all had different activity profiles across the strains. Of the control strains (ATCC 27853 and PAO1), gentamicin had the greatest inhibitory action (1–2 µg/mL) (1.7–3.5 µM), followed by Ag-tdda-phen (8–32 µg/mL) (6.6–26.6 µM), with the least active being Cu-tdda-phen, and Mn-tdda-phen (both had the same activity of ca. 16 and 32 µg/mL) (ca. 21 and 43 µM). In contrast to these control strains, the metal-tdda-phen complexes were active against all of the CF clinical isolates. Against CF1, these three metal complexes had activity (8–16 µg/mL) (10.9–13.3 µM) comparable to that of gentamicin, and were more active than this antibiotic against CF2 (8–16 µg/mL) (10.9–13.3 µM) and CF3 (64–128 µg/mL) (53.2–174 µM). Following the MIC measurement, it was only the three metal-tdda-phen complexes and gentamicin that were used in further biological assays.

### 2.2. Metal-Tdda-Phen Complexes Are Able to Inhibit Biofilm Formation and Disrupt Established Biofilm

Although bacteria are unicellular organisms, they exist in natural and clinical settings predominately within biofilms. To determine the capacity of the test complexes to prevent the formation of biofilms by *P. aeruginosa* isolates, bacterial cells were pre-treated with the three metal-tdda-phen complexes or gentamicin for 48 h. Similarly, the post-treatment of established mature biofilms (at 48 h) was also carried out to compare the effect of the compounds on developing or developed biofilms. For both pre- and post-treatment studies, cellular viability (resazurin stain) and biofilm biomass (crystal violet stain) were measured. Both stains could be used sequentially on the same treated biofilm sample, as resazurin measures only viable cells, whereby crystal violet binds to the extracellular matrix of biofilms and stains all cells present in the biomass, with no discrimination between live or dead cells. However, one limitation of the resazurin method is that it does not allow for the detection of persister cells that may be present in the biofilm, and therefore was measured just to accompany biofilm biomass data.

The pre-treatment of potential biofilm-forming bacterial cells with the metal-tdda-phen complexes and gentamicin was dose-dependent for both cellular viability (Figure 1) and biofilm biomass (Figure 2), inhibiting biofilm formation at concentrations lower than the concentration required to restrict planktonic bacterial growth, as previously outlined (Table 1), and with contrasting activity between the laboratory strains and CF clinical isolates. Gentamicin was found to inhibit biofilm formation in strains ATCC 27853 and PAO1, upwards of 50% at a concentration of 1.7 µg/mL (2.9 µM) and 1.9 µg/mL (3.3 µM), respectively, while it required 8.6–39.2 µg/mL (7.2–53.3 µM) of the metal-tdda-phen complexes to illicit the same response. The activity of Mn-tdda-phen, Cu-tdda-phen and Ag-tdda-phen varied across the clinical isolates, and were either comparable (CF1) or superior to gentamicin (CF2 and CF3). For CF1, the biomass was inhibited 50% by Mn-tdda-phen at a concentration of 4 µg/mL (5.4 µM), followed by gentamicin (4.2 µg/mL, 7.3 µM), Ag-tdda-phen (6.1 µg/mL, 5.1 µM) and Cu-tdda-phen (8.4 µg/mL, 11.3 µM). To inhibit biofilm formation by 50% in CF2, 2.5 µg/mL (3.4 µM) of Mn-tdda-phen, 4.4 µg/mL (5.9 µM) of Cu-tdda-phen, 6.2 µg/mL (5.2 µM) of Ag-tdda-phen and 81.3 µg/mL (141.2 µM) of gentamicin were required, respectively, while for CF3, which formed the strongest biofilm, 26.5 µg/mL (35.6 µM) of Cu-tdda-phen, 29.7 µg/mL (40.4 µM) of Mn-tdda-phen, 33.4 µg/mL (27.8 µM) of Ag-tdda-phen and 116.5 µg/mL (202.4 µM) of gentamicin were required to inhibit the biofilm formation by 50%. Interestingly, pre-treatment of the clinical isolates with sub-MICs of gentamicin increased the biomass, relative to the untreated sample, which has previously been observed [26].

As antibiotics are predominantly administered in clinical practice to counteract an existing infection, evaluating the effects on established mature biofilms (post-treatment) provided a more pragmatic insight to the anti-biofilm activity of the test complexes. None of the test complexes caused the complete eradication of *P. aeruginosa* biofilms at the concentrations presented by cellular viability (Figure 1) and biofilm biomass (Figure 2). The activity of gentamicin diminished considerably across all strains when administered to established biofilms, a feature which has commonly been reported [27]. While gentamicin maintained moderate activity against ATCC 27853, with the minimum biofilm eradication concentration (MBEC) removing 50% of the biomass (MBEC_50_) being 19.6 µg/mL (34.1 µM), the MBEC_50_ for PAO1 was considerable larger (181 µg/mL, 314 µM). At the highest concentration of gentamicin examined (256 µg/mL, 445 µM) against the clinical isolates CF1, CF2 and CF3, there was still approximately 52%, 67% and 98% biofilm remaining, respectively, relative to the untreated samples. It is well known that, as a mature biofilm is established, the extracellular matrix slows or prevents the penetration of antibiotics into the microbial communities embedded in it. Surprisingly, against the clinical isolates, the metal-tdda-phen complexes all showed similar activity to each other (MBEC_50_ range ca. 12–51 µg/mL, 14–70 µM), and were superior to gentamicin.

### 2.3. Metal-Tdda-Phen Complexes Enhanced the Anti-Biofilm Activity of Gentamicin

Biofilm-associated tolerance is multifactorial and is attributed to restricted penetration of the antimicrobial agent and the different metabolic states and virulence factors of the microbial community within the biofilm. Current research approaches aimed at increasing the likelihood of biofilm eradication include the use of non-antibiotic complexes, either alone or in combination with conventional antibiotics [28]. The combination of metal-tdda-phen complexes (0–64 µg/mL) with gentamicin (0–32 µg/mL) was tested on established mature (48 h) biofilms of *P. aeruginosa* using the checkerboard assay, and the results are presented in Figure 3 for all strains; ATCC 27853 (A–C), PAO1 (D–F), CF1 (G–I), CF2 (J–L) and CF3 (M–O). To varying degrees, the addition of the metal-tdda-phen complexes enhanced the anti-biofilm activity of gentamicin across all strains. At the highest concentration of gentamicin alone (32 µg/mL, 55.5 µM), ca. 19–37% of the biofilm was removed from reference strains ATCC 27853 and PAO1, whilst the same concentration of gentamicin, combined with 4 µg/mL (3.3–5.4 µM) of the metal-tdda-phen complexes, was able to degrade over 50% of the biofilm. Increasing the concentration of the added metal-tdda-phen complexes prompted the further disintegration of the established biofilm. Most noteworthy, is the combination of the antibiotic with Mn-tdda-phen and Ag-tdda-phen at 64 µg/mL (53.2 and 87 µM, respectively), which removed 99.6% and 99.1% of PAO1 biofilm, respectively. Enhanced activity was also maintained across the clinical isolates, with the combined metal-tdda-phen/gentamicin combinations presenting a dose-dependent response. At the highest test concentration, the combination of metal-tdda-phen complexes (64 µg/mL, 53.2–87 µM) with gentamicin (32 µg/mL, 55.5 µM) dismantled over 90% of the mature biofilm for all three metal-tdda-phen complexes in CF2 (93.5–96.5% dismantled) and CF3 (93.9–99.6%), while in CF1, the combination results were: Ag-tdda-phen (92.8%), Cu-tdda-phen (86.1%), Mn-tdda-phen (83.1%).

The fractional inhibitory concentration (FIC) index (FIC_I_) for combinations of metal-tdda-phen complexes with gentamicin was calculated according to the equation: FIC_I_ = FIC_(metal-tdda-phen complex)_ + FIC_(gentamicin)_. Results for the interactions that occurred between the metal-tdda-phen complexes and gentamicin combinations in the selected *P. aeruginosa* strains, PAO1 and CF3, are given in Table 2. The data revealed synergistic activity (FIC ≤ 0.5) against the established mature biofilms for all of the metal-tdda-phen/gentamicin combinations.

### 2.4. Metal-Tdda-Phen Complexes Reduce Components in a Biofilm; This Activity Is Enhanced When Combined with Gentamicin

In biofilms, microbes are held together by a protective polymeric extracellular matrix (ECM) comprised of polysaccharides and extracellular DNA (eDNA), and also virulence factors such as pyocyanin and pyoverdine. To understand how the metal-tdda-phen complexes effect both the formation of biofilms (pre-treatment) and the reduction of established mature biofilms (post-treatment), MBIC_50_ and MBEC_50_ doses were applied to *P. aeruginosa* strains PAO1 and CF3, alone and in the presence of gentamicin, and the results are presented in Figure 4.

Exopolysaccharide is an important component of the ECM for the formation and stabilisation of the biofilm structure, and therefore the reduction of this component would be a beneficial target for antimicrobial agents. The pre-treatment of PAO1 and CF3 with metal-tdda-phen complexes caused an attenuation in exopolysaccharides. Exposure to Mn-tdda-phen caused the greatest reduction of exopolysaccharide in PAO1 (74% ± 4), followed by Cu-tdda-phen (67% ± 1), Ag-tdda-phen (49% ± 6) and gentamicin (47% ± 9). Combined pre-treatment of Mn-tdda-phen, Cu-tdda-phen and Ag-tdda-phen with gentamicin reduced exopolysaccharides in PAO1 by 64% ± 6, 72% ± 5 and 81% ± 5, respectively, reducing the efficacy of Mn-tdda-phen as a single agent, but enhancing the effect of the other two metal-tdda-phen complexes. Pre-treatment of the clinical isolate CF3 with the test agents saw a reduction in the activity of metal-tdda-phen complexes (52–55% ± 7) and gentamicin (38% ± 6) against the exopolysaccharides, compared to that seen for PAO1. However, in contrast to PAO1, the Mn-tdda-phen/gentamicin combination was the most effective (77% ± 1) at reducing exopolysaccharide in the clinical isolate. For the post-treatment studies, the biofilms were allowed to grow for 48 h and then exposed to the test complexes for a further 24 h. For PAO1, the reduction of exopolysaccharides by the metal-tdda-phen complexes (41–46% ± 9) was similar to that observed for gentamicin (44% ± 9), while the combined treatment again intensified the activity, prompting a 73–78% reduction in exopolysaccharide. For the treatment of CF3, gentamicin was the least effective and Ag-tdda-phen the most when used singly (latter complex reduced the exopolysaccharide in the biofilm by 61% ± 9). For the metal-tdda-phen/gentamicin combinations, activity increased somewhat uniformly (72–77% reduction).

Extracellular DNA (eDNA) is recognised as an essential constituent of the ECM. It facilitates biofilm formation and enhances biofilm strength and stability through the bridging by divalent ions to other components in the matrix. It has also been observed that eDNA increases the tolerance of *P. aeruginosa* biofilms toward cationic antibiotics, such as aminoglycosides, by binding directly to the antibiotic and preventing penetration [29]. The pre-treatment of PAO1 and CF3 with metal-tdda-phen complexes or gentamicin singly did not cause any appreciable difference in eDNA content when compared to the untreated sample. The combination treatments were unilaterally more effective, with Ag-tdda-phen/gentamicin inhibiting eDNA in PAO1 by 50% ± 11, Cu-tdda-phen/gentamicin by 49% ± 4 and Mn-tdda-phen/gentamicin 34% ± 6. A similar trend was observed for CF3. With the exception of gentamicin, the post-treatment of established biofilms of both strains with single metal-tdda-phen complexes was more effective than pre-treatment. A dramatic improvement in eDNA inhibition was witnessed when both of the established biofilms were exposed to all of the metal-tdda-phen/gentamicin combinations (66–75% reduction), with Cu-tdda-phen/gentamicin being the most aggressive.

The prevalence and progression of infection by *P. aeruginosa* in the host is dependent upon the secretion of numerous extracellular molecules, such as the virulent factors, pyocyanin and pyoverdine. Pyocyanin is one of the major components dictating the advancement of infection and biofilm formation, especially in CF patients, and is under quorum sensing (QS) control. It is redox active and promotes eDNA release in *P. aeruginosa* by inducing cell lysis mediated via hydrogen peroxide production. The extent of pyocyanin production varied amongst the strains of *P. aeruginosa* tested, untreated PAO1 manufactured about 10 µM of pyocyanin, while the untreated CF clinical isolate, CF3, produced upwards of 80 µM. The results of the pyocyanin assay are presented as a percentage of production relative to the untreated sample for comparison (Figure 4). Pre-treatment with the Ag-tdda-phen/gentamicin combination effected the greatest reduction in pyocyanin in both PAO1 (80% ± 9) and CF3 (65% ± 6). The manganese and copper metal-tdda-phen/gentamicin combinations also reduced pyocyanin production when compared to their treatment as a single agent. Surprisingly, the pre-treatment of PAO1 with Cu-tdda-phen alone actually enhanced the production of pyocyanin (106% ± 8), and gentamicin also had a similar positive influence on CF3 (116% ± 13). In general, the single test compounds and their gentamicin combinations were much less effective at reducing pyocyanin production by the two mature biofilms. Although Ag-tdda-phen alone increased pyocyanin in PAO1 (134%), it was the Ag-tdda-phen/gentamicin combination that caused the greatest reduction of pyocyanin (35% ± 9 in PAO1 and 49% ± 7 in CF3).

Pyoverdine, the main siderophore of *P. aeruginosa*, has a high affinity for ferric (Fe^3+^) iron and thus is responsible for obtaining extracellular iron, which is an essential nutrient for biofilm formation [30]. In addition, pyoverdine can regulate the production of multiple bacterial virulence factors. In the pre-treatment studies (Figure 4), it was found that only the Ag-tdda-phen/gentamicin caused a notable reduction in pyoverdine production (by 16% ± 5 in PAO1 and 49% ± 6 in CF3), with most of the remaining test substances actually enhancing production (by up to 134%). Post-treatment of the two established biofilms with the single metal-tdda-phen compounds had little or no effect on pyoverdine production with respect to the control. Whilst gentamicin singly and the Mn-tdda-phen/gentamicin combination caused a sizable reduction in pyoverdine in the reference strain (24% ± 8 and 26% ± 9, respectively) and the clinical isolate (30% ± 8 and 41% ± 9, respectively), Cu-tdda-phen/gentamicin and Ag-tdda-phen/gentamicin were only effective against CF3 (41% ± 9 and 48% ± 7 reduction, respectively).

In addition to the components of the biofilm, the liquid culture they were grown in was aspirated, and the bacterial cells present were assessed through the plate count method. As presented in Figure 5, untreated PAO1 had a 7.6 × 10^8^ colony forming unit (CFU)/mL, while pre-treatment with metal-tdda-phen complexes saw a slight reduction of 7.1–5.2 × 10^8^ CFU/mL, and gentamicin had 3.9 × 10^8^ CFU/mL. Complementing the biofilm component data, the combination of both the metal-tdda-phen complexes with gentamicin saw the greatest reduction of viable cells, with Ag-tdda-phen demonstrating a one-fold reduction, to 7.9 × 10^7^ CFU/mL. Pre-treatment of CF3 with gentamicin reduced the viable cells from 7.5 × 10^8^ CFU/mL to 7 × 10^8^ CFU/mL, followed by Cu-tdda-phen (5.9 × 10^8^ CFU/mL), Ag-tdda-phen (5.5 × 10^8^ CFU/mL) and Mn-tdda-phen (5.4 × 10^8^ CFU/mL). Akin to PAO1, metal-tdda-phen/gentamicin reduced viable cells in the culture to 3.3 × 10^8^ CFU/mL–2.2 × 10^8^ CFU/mL. Post-treatment of PAO1 and CF3 had similar trends, metal-tdda-phen complexes as singular agents slightly outperforming gentamicin. However, the combination of Mn-tdda-phen/gentamicin and Cu-tdda-phen/gentamicin saw a 2-fold reduction in viable cells and Ag-tdda-phen/gentamicin a 3-fold reduction.

## 3. Discussion

*Pseudomonas aeruginosa* is the most prevalent pathogen in the lungs of cystic fibrosis (CF) patients and the leading cause of morbidity and mortality [12]. Ireland has the highest rate of CF per capita in the world, with approximately 7 in every 10,000 people living with the disease; almost three times the average rate in other European countries and the United States [31]. Conventional antibiotic treatment strategies for *P. aeruginosa* in CF patients are only partially effective and contribute to antibiotic resistance. *P. aeruginosa* predominantly reside in biofilms in the lungs of CF patients, which are also up to 1000 times more resistant to antibiotic treatment than their planktonic counterparts. Antibiotic treatment strategies are still being trialled and optimized for CF patients, and there have been numerous studies comparing mono- and combination-antibiotic therapies [32,33]. Amongst the many alternative therapeutic approaches documented are the guluronate polymer OligoG (derived from alginate), a simple thiol cysteamine, a novel synthetic peptide, nitric oxide and gallium(III) compounds [34]. Ga^3+^ metal ions have been shown to disrupt biofilm formation and protect against *P. aeruginosa* infections in vitro in human serum [35], mouse models [36,37] and humans [38]. Moreover, there is a decreased likelihood that *P. aeruginosa* can develop resistance to gallium compared to the classic small molecule antibiotics [39]. Additional studies showed that co-encapsulation of Ga^3+^ with gentamicin in liposomes was more effective than gentamicin alone for eradicating *P. aeruginosa* in both planktonic and biofilm forms [40]. There is clearly scope for the development of novel metal complexes, either acting alone or in combination with antibiotics, as a new therapeutic regime for *P. aeruginosa* infections in CF patients.

We have investigated the anti-biofilm capabilities of copper(II), manganese(II), and silver(I) complexes containing 1,10-phenanthroline (phen) and 3,6,9-trioxaundecanedioate (tddaH_2_) ligands, for their ability to inhibit the planktonic growth of *P. aeruginosa* strains isolated from the lungs of CF patients in Irish hospitals. These metal complexes have previously shown anti-bacterial and anti-fungal capabilities, and the inclusion of the phen ligand is key to their antimicrobial properties. Furthermore, the tdda dianionic ligand greatly enhances the water solubility of the complex. The results presented in Table 1 clearly demonstrate the superior potency of the metal-tdda-phen complexes when compared to the anti-*P. aeruginosa* activities of the metal-free phen, tddaH_2_ and the simple copper(II), manganese(II) and silver(I) salts (essentially free metal ions). The antibiotic, gentamicin, was the most active compound against the reference strains (ATCC 27853 and PAO1). Whereas the efficacy of gentamicin dampened across the clinical isolates (CF1–CF3), the metal-tdda-phen complexes maintained clinically relevant activity. Taking the mean minimum inhibitory concentration (MIC) for each bacterial strain, the order of planktonic growth inhibition by the metal-tdda-phen complexes was: Cu-tdda-phen > Ag-tdda-phen > Mn-tdda-phen. Gandra et al. [17] reported that the Mn-tdda-phen complex was extremely well tolerated by the mammalian lung cell line, A549, with a concentration capable of reducing cellular viability by 50% (CC_50_) of 261.67 mg/L. In the present study, the MIC range for Mn-tdda-phen was 8–128 µg/mL, suggesting that this complex would be significantly selective for the CF1 and CF2 clinical strains of *P. aeruginosa*. Furthermore, at the highest concentration of Mn-tdda-phen required to inhibit CF3 growth, the complex would still have moderate selectivity.

The biofilm lifecycle of *P. aeruginosa* is an important virulence attribute for establishing infections in CF patients. All of the metal-tdda-phen complexes were able to inhibit biofilm formation (pre-treatment) in terms of both biofilm biomass and cellular viability, outperforming gentamicin across the three clinical isolates. Moreover, all three of the metal-tdda-phen complexes were able to disrupt the established mature biofilm (post-treatment) of the *P. aeruginosa* strains in a dose-dependent manner. Similar activity has previously been reported by Viganor et al. [18] for the related complex cations, [Ag(phendione)_2_]^+1^ and [Cu(phendione)_3_]^+2^ (phendione = 1,10-phenanthroline-5,6-dione), against 32 Brazilian clinical isolates of *P. aeruginosa*. Moreover, the same phendione complexes were shown to exhibit anti-biofilm action against carbapenemase-producing *Acinetobacter baumannii*, another multi-drug resistant (MDR) Gram-negative bacterium [23]. Presently, the MIC of Mn-tdda-phen for CF3 was 128 µg/mL (174 µM), and when an established biofilm of this strain is treated at the same concentration, it is reduced by ~79%. The MIC of both Cu-tdda-phen and Ag-tdda-phen was 64 µg/mL (86 and 53.2 µM, respectively) for CF3, and the reduction in biofilm biomass after treatment at this concentration was ~70% and ~51%, respectively. These anti-biofilm studies indicated that there is potential for enhanced anti-biofilm activity, through a combination of the metal-tdda-phen complexes with gentamicin. At the highest concentration of the metal-tdda-phen complexes (64 µg/mL) in combination with gentamicin (32 µg/mL), there was >90% biofilm removal for CF2 and CF3 and >83% for CF1, and the fractional inhibitory concentration (FIC) index showed a positive synergism for all three metal-tdda-phen complexes in the presence of gentamicin. A recent clinical study in Ireland on cystic fibrosis *P. aeruginosa* clinical isolates with the same laboratory reference strains (ATCC 27853 and PAO1) showed that tobramycin (aminoglycoside) and ceftazidime (cephalosporin) was the only effective combination to show positive synergism (FIC ≤ 0.5) against a mature biofilm of a single clinical isolate at a concentration of 64 mg/L [41]. However, as mentioned earlier, the evolving resistance to antibiotics is problematic and of concern when considering combinations of established antibiotics for the treatment of *P. aeruginosa* infections in CF patients. Novel therapeutics, where resistance is less likely to be a factor, would be favourable for future clinical interventions.

Since significant activity was evident for the metal-tdda-phen complexes, acting either alone or in combination with gentamicin, against biofilms formed by the PAO1 reference strain and the CF3 clinical isolate, individual components of the biofilm were tested to see if there was a specific molecular target. These targets included exopolysaccharide and extracellular DNA (eDNA), which are two key components of the polymeric extracellular matrix (ECM), and the virulence factors, pyocyanin and pyoverdine, which are found in the ECM. Many *P. aeruginosa* strains are capable of synthesizing the three exopolysaccharides, alginate, Psl, and Pel, which play an important role in biofilm formation and the stabilization of its structure. Strains isolated from CF patients are commonly found to over-produce alginate, and biofilms of these strains have been observed to have a much higher resistance to aminoglycosides, such as gentamicin, than their isogenic non-mucoid strain [42]. Additional studies have also implicated both Psl [43] and Pel [44] in the biofilm-associated tolerance towards aminoglycosides, thus making exopolysaccharides an important component of the ECM. It is thought that Psl associates with eDNA through hydrogen bonding to form eDNA–Psl fibres in the centre of pellicles [45], and cationic Pel cross-links eDNA in the biofilm stalk via ionic interactions [46]. eDNA is known to chelate divalent metal cations, such as Mg^2+^, Mn^2+^, Zn^2+^ and Ca^2+^ [47], with the latter creating strong electrostatic interactions which give the biofilm integrity [48], and there have been biofilm studies conducted on the use of metal chelators, such as ethylenediaminetetraacetic acid (EDTA). In this context, EDTA alone was found to be 1000-fold more destructive to a PAO1 biofilm than gentamicin and, furthermore, a combination of EDTA (50 mM) with gentamicin (50 μg/mL) caused a complete reduction of biofilm viability (>99%) and detachment of cells [49]. The present study demonstrated that the enhanced anti-biofilm activity of metal-tdda-phen/gentamicin combinations was associated with a reduction in both exopolysaccharide and eDNA. It has been widely reported that the treatment of *P. aeruginosa* biofilms with DNase I, an enzyme that cleaves DNA, significantly disrupts biofilms and enhances antibiotic penetration [50]. As CF sputum is rich in eDNA (<1–20 mg/mL), this is thus a desirable molecular target. The copper(I) complex cation, [Cu(phen)_2_]^+^, in the presence of reducing agents, was the first reported example of a novel artificial metallonuclease possessing potent DNA cleavage capability [51]. Our group has previously shown that copper(II)-dicarboxylate-phen complexes also display effective nuclease capability, even in the absence of added reductant [52]. In this context, treating a mature biofilm with Cu-tdda-phen prompted the greatest decrease in eDNA in both the reference and clinical isolates. Our data suggest that the metal-tdda-phen complexes cause biofilm destabilization by interacting with extracellular DNA and exopolysaccharide in the ECM, possibly through Psl. However, further studies will be required to establish the role of Psl and Pel in these interactions. This destabilization could allow the previously inhibited gentamicin into the biofilm to enhance its cytotoxic action, which has been further supported by the viable cell counts.

Pyocyanin and pyoverdine are both virulence factors and are carefully controlled by quorum sensing networks. Pyocyanin is a metabolite involved in redox reactions in the biofilm, and pyoveridine is the main siderophore of *P. aeruginosa* involved in iron metabolism. Both virulence factors were reduced upon treatment with metal-tdda-phen/gentamicin combinations, but to a lesser extent than the exopolysaccharide and eDNA. The encouraging results obtained from this study prompt further investigations into more fully elucidating the exact mechanism(s) of action of the metal-tdda-phen complexes and their gentamicin combinations. Furthermore, investigations of combinations of the metal complexes with other antibiotics are also warranted.

## 4. Materials and Methods

### 4.1. Test Complexes

The metal complexes used for this study were originally synthesised by McCann et al. [53] as follows: [Cu(3,6,9-tdda)]·H_2_O (Cu-tdda) and {[Cu(3,6,9-tdda)(phen)_2_]·3H_2_O·EtOH}_n_ (Cu-tdda-phen); [Mn(3,6,9-tdda)(H_2_O)_2_]·2H_2_O (Mn-tdda) and {[Mn(3,6,9-tdda)(phen)_2_]·3H_2_O·EtOH}_n_ (Mn-tdda-phen) and [Ag_2_(3,6,9-tdda]·2H_2_O (Ag-tdda) and [Ag_2_(3,6,9-tdda)(phen)_4_]·EtOH (Ag-tdda-phen). The metal complexes were prepared in accordance with those described by Gandra et al. [17].

In order to establish that the observed biological effects were due to the discrete complexes rather than their free ligands, 1,10-phenathroline (phen) and 3,6,9-trioxaundecanedioic acid (tddaH_2_), and metal salts, manganese chloride (MnCl_2_), copper chloride (CuCl_2_), and silver nitrate (AgNO_3_) were also assessed (all chemicals were purchased from the Sigma-Aldrich company (St. Louis, MO, USA) and used without further purification). The aminoglycoside antibiotic, gentamicin (Sigma-Aldrich) was also incorporated into the study, as it is an antibacterial agent commonly used to treat *P. aeruginosa* in infected CF patients. All stock complexes (10 mg/mL) were diluted in tryptic soy broth (TSB) media for desired test concentrations.

### 4.2. Bacterial Strains and Culture Conditions

Clinical isolates of *Pseudomonas aeruginosa* (CF1, CF2 and CF3), from cystic fibrosis patients at local Irish hospitals, along with standard laboratory strains ATCC 27853 and PAO1, were obtained from Dr. Gordon Cooke, Department of Science TU Dublin—Tallaght campus. ATCC 27853 originated from a blood culture in Boston, USA [54] while PAO1 originated form a burn wound in Melbourne, Australia [55]. Stock frozen cultures were maintained at −80 °C in 20% (*v*/*v*) glycerol solutions in tryptic soy broth (TSB; Lab M). Frozen cultures were streaked onto plates of tryptic soy agar (TSA; Lab M) and incubated overnight at 37 °C. A single colony of bacteria was then inoculated into TSB (50 mL) to prepare the bacterial inoculate for the experiments (37 °C at 200 rpm). Bacterial cell suspensions of 10^6^ CFU/mL were used for all subsequent experiments.

### 4.3. Antimicrobial Susceptibility Testing of Complexes on Pseudomonas aeruginosa Strains

#### 4.3.1. Effects of Test Compounds on Planktonic Bacteria

All metal complexes and controls were tested for their activity on the clinical isolates (CF1–CF3) and laboratory strains (ATCC 27853 and PAO1) of *P. aeruginosa*, using the standard broth micro–dilution method to establish their minimum inhibitory concentration (MIC). The test complexes were two-fold serially diluted and mixed with equal volumes (100 µL) of diluted bacteria in 96-well plates (Cruinn), thus making a final concentration range of the complexes tested, and included the antibiotic control gentamicin, between 0.5 and 256 µg/mL. The MIC measurements were recorded on the basis of turbidity according to EUCAST guidelines [56], after 16–18 h of incubation at 37 °C with agitation.

#### 4.3.2. Effects of Test Compounds on Biofilm Formation (Pre-Treatment)

The activity of all complexes and controls on biofilm formation of all *P. aeruginosa* strains were tested by measuring cellular viability and the total biomass in the cell culture wells. For both assays, 100 µL of the bacterial culture (made up in TSB medium) was distributed into each well of flat-bottomed 96–well polystyrene microtitre plates (Cruinn) and incubated for 48 h at 37 °C in the presence of all test compounds (100 µL), administered at a concentration range of 0.5–256 µg/mL. The supernatants from each well were removed, and the wells were then washed three times with phosphate buffered saline (PBS). Viable cells in the biofilm were assessed by a fluorometric assay that measures the metabolic capacity of cells [57]. TSB (250 µL) containing 5% (*v*/*v*) resazurin (Sigma-Aldrich) was added to the plates and incubated for 20 min at 37 °C. Fluorescence was then measured at λexcitation = 560 nm and λemission = 590 nm, with a Varioskan LUX (Thermo Scientific, Waltham, MA, USA) microplate reader. Then, the resazurin stain was removed and replaced by 250 µL of 0.1% (*w*/*v*) crystal violet solution on the same plate [58] to evaluate biofilm biomass. After 20 min, the crystal violet stain was removed, and the wells werewashed three times. The wells were dried and 250 µL of 30% acetic acid (Sigma–Aldrich) was added to release the bound crystal violet. Absorbance was measured on a Multiskan GO (Thermo Scientific, Waltham, MA, USA) microplate reader at 590 nm.

#### 4.3.3. Effects of Test Compounds on the Mature Biofilm (Post-Treatment)

In addition to testing the effects of the metal-tdda-phen complexes on biofilm formation (pre-treatment) of the *P. aeruginosa* strains, their ability to weaken an established biofilm was also assessed with the post-treatment of mature biofilms. Moreover, 100 µL of the bacteria culture (in TSB) was added to each well of flat bottomed 96–well polystyrene microtitre plates (Cruinn) and incubated for 48 h at 37 °C, to allow for biofilm formation. After 48 h incubation, the test complexes and control were added over a range of concentrations (0.5–256 µg/mL) to mature biofilms and incubated for an additional 24 h. To the untreated control, 100 µL of fresh media was added to the wells. Cellular viability and biofilm biomass were measured using resazurin staining and crystal violet staining, as previously described in Section 4.3.2.

#### 4.3.4. Checkerboard Assay for Mature Biofilms

The anti-biofilm activity of individual metal-tdda-phen complexes alone and in combination with gentamicin were evaluated against the *P. aeruginosa* strains using the broth micro-dilution checkerboard technique [59]. Mature biofilms were prepared as previously described in Section 4.3.3. The preformed biofilms in the wells of separate 96-well microtitre plates were rinsed with PBS and 100 µL of the metal-tdda-phen complexes and gentamicin (two-fold serial dilutions); each was added to the wells containing the biofilms. Following incubation for 24 h at 37 °C, the biofilm biomass and cell viability were measured as previously described.

The fractional inhibitory concentration (FIC) index (FIC_I_) for combinations of testing agents was calculated for selected *P. aeruginosa* strains PAO1 and CF3, according to the equation: FIC_I_ = FIC_(metal-tdda-phen complex)_ + FIC_(gentamicin)._ Where, FIC_(metal-tdda-phen complex)_ = (MBEC* of metal-tdda-phen complex in combination with gentamicin)/(MBEC* of metal-tdda-phen complex alone) and FIC_(gentamicin)_ = (MBEC* of gentamicin in combination with metal-tdda-phen complex)/(MBEC* of gentamicin alone). These strains were selected due to their contrasting activity profiles observed in the previous tests, and because they formed the strongest biofilms (evidenced by crystal violet staining). The following criteria were used to interpret the FIC_I_: ≤0.5 synergy; 0.5–4.0 indifferent; >4.0 antagonism. *The minimum biofilm eradication concentration (MBEC) was defined as the minimal concentration of the compound required to eradicate the biofilm.

### 4.4. Effects of Test Complexes on Individual Components of Biofilm

Selected *P. aeruginosa* strains, PAO1 and CF3, were incubated in the presence and absence of metal-tdda-phen complexes, alone and in combination with gentamicin. To assess the effects of biofilm formation (pre-treatment), experiments were set up according to Section 4.3.2 with strains pre-treated to 0.5 × minimum biofilm inhibitory concentration (MBIC). To assess the effects on established mature biofilms (post-treatment), experiments were set up according to Section 4.3.3, with biofilms grown to 48 h, then treated with 0.5 × minimum biofilm eradication concentration (MBEC) for a further 24 h. The bacterial cultures were harvested and the subsequent components, including exopolysaccharide, extracellular DNA (eDNA), pyocyanin and pyoverdine, which were analysed by the different methods described below.

#### 4.4.1. Extraction and Quantification of Exopolysaccharide

Extraction and quantification of exopolysaccharide were performed by using a modified protocol of Tribedi and Sil [60]. Biofilms were grown on a glass surface and then extracted by gentle dislodgment using a cell scrapper (25 cm, Sarstedt) and PBS. The biofilm suspension was centrifuged at 4500× *g* for 15 min and the supernatant was collected. The resulting pellet was treated with 10 mM/L EDTA, vortexed and centrifuged again to extract cell-bound polysaccharide. The supernatant was harvested and mixed with the first harvested supernatant. An equal volume of chilled absolute ethanol was added to the pooled supernatant and incubated at −20 °C for 1 h. After centrifugation at 10,000× *g* for 10 min, the pellet was resuspended in sterile water and measured by the phenol-sulphuric acid extraction method [61]. Briefly, 1 mL of the precipitated exopolysaccharide solution was mixed with an equal amount of chilled phenol (5%) and 5 mL of concentrated sulphuric acid. The resulting red colour was measured at 490 nm on a spectrophotometer (Hach, Bio-Sciences).

#### 4.4.2. Extraction and Quantification of Extracellular DNA (eDNA)

Extracellular DNA (eDNA), a key component of the biofilm, was extracted from the matrix using a modified enzymatic method, as reported by Wu and Xi [62]. To release the eDNA from the matrix, the biofilms were first exposed to cellulase (10 µg/mL, Novozymes) and α-amylase (10 µg/mL, Sigma-Aldrich) to hydrolyse the polysaccharides, and incubated at 50 °C for 30 min [63,64]. Proteinase K (5 µg/mL, Sigma-Aldrich) was then added to the mixture for a further 30 min, to degrade proteins, and the supernatant was sequentially filtered through a 0.45 µm and then a 0.20 µm filter to remove residual bacteria. The eDNA was precipitated by 1 volume of cetyltrimethylammonium bromide (CTAB, Sigma-Aldrich) solution (1% CTAB in 50 mM Tris–10 mM EDTA, pH 8.0), incubated at 65 °C for 30 min, followed by centrifugation at 5000× *g* for 10 min. Each pellet was resuspended in high-salt TE buffer (10 mM Tris-HCL, 0.1 mM EDTA, 1 M NaCl, pH 8.0) and 0.7 volume of cold isopropanol, and incubated at −20 °C for 3 h. After centrifugation at 10,000× *g* for 15 min, pellets were suspended in TE buffer (10 mM Tris-HCL, 0.1 mM EDTA, pH 8.0) and an equal volume of equilibrated phenol:chloroform:isoamyl alcohol (25:24:1) and centrifuged again. The upper aqueous layer was transferred to a tube with an equal volume of chloroform:isoamyl alcohol (24:1) and centrifuged again. The supernatant was then precipitated with 2 volumes of cold ethanol (70%) and one tenth the final volume of sodium acetate (0.3 M), incubated at −20 °C for 1 h and centrifuged at 10,000× *g* for 15 min. The pellets were finally washed twice with ethanol and resuspended in MilliQ water. The purity of the DNA was checked by determining the ratio of absorbance at 260 nm and 280 nm (A260/280) using a MultiskanTM GO (Thermo Scientific, Waltham, MA, USA) UV spectrophotometer with the µDrop plate, and the DNA was quantified fluorometrically (Qubit 3.0 Fluorometer, Invitrogen, Carlsbad, CA, USA) by using SYBR Green I (Molecular Probes), as described by Zipper [65]. The size of extracellular DNA was analyzed on a 1% (*w*/*v*) agarose gel in a Tris-Borate-EDTA (TBE) buffer.

#### 4.4.3. Quantification of Pyocyanin

Pyocyanin is a toxic, QS-controlled metabolite produced by *P. aeruginosa* enabling biofilm formation. After a 48 h incubation, the bacterial strains were centrifuged at 4500× *g* for 20 min and the cell-free supernatant was used. Pyocyanin from the supernatant was extracted and measured by the methods of Essar et al. [66]. Then, 5 mL of filtered culture supernatant was extracted with chloroform (3 mL), vortexed and centrifuged at 4500× *g* for 15 min. The chloroform phase was transferred to a fresh tube with 1 mL of 0.2 M hydrochloric acid (HCL). After centrifugation at 4500× *g* for 4 min, the top layer was collected and its absorption was measured on a spectrophotometer (Hach, Bio-Sciences) at 520 nm. Pyocyanin concentrations, expressed as micrograms per millilitre (µg/mL) of culture supernatant, were determined by multiplying the OD at 520 nm by 17.072 (extinction coefficient).

#### 4.4.4. Quantification of Pyoverdine

Pyoverdine is the main siderophore of *P. aeruginosa*, deployed in severely iron-limited environments, to assure sufficient supply of this essential nutrient, and is found at increased concentrations in biofilms. After a 48 h incubation, the bacterial strains were centrifuged at 4500× *g* for 20 min and the cell-free supernatant was used to measure pyoverdine. The relative concentration of pyoverdine in all treated supernatant with respect to control was measured through florescence at an excitation wavelength of 405 nm and an emission wavelength of 465 nm [67], with a Varioskan LUX (Thermo Scientific, Waltham, MA, USA) microplate reader.

#### 4.4.5. Quantification of Bacteria

In addition to the components of the biofilm, liquid medium with bacteria was collected and centrifuged (4500× *g* for 20 min), after which the supernatant was discarded, and the pellet was resuspended in 1 mL PBS. Ten-fold serial dilutions of the resuspended bacteria were then prepared and plated on nutrient agar utilising the Miles and Misra (20 μL drop) method. Agar plates were incubated at 37 °C for 16–18 h, after which colony forming unit (CFU) per mL was calculated using the formula: CFU/mL = average number of colonies for a dilution × 50 × dilution factor.

### 4.5. Statistics

All experiments were performed in triplicate, in three independent experimental sets. All data were statistically analysed using one-way analysis variance (ANOVA). The data are expressed as mean values with standard deviations (SDs). Data with *p*-values smaller than 0.05 were considered statistically significant. All statistical analyses were performed using GraphPad Prism (GraphPad Software Inc., San Diego, CA, USA).

## 5. Conclusions

Cu(II)-, Mn(II)- and Ag(I)-tdda-phen chelate complexes enhance the effects of gentamicin for eradicating clinical *P. aeruginosa*-established biofilms, originating from CF patients with individual response and resistance profiles, and therefore offer an alternative combination therapeutic approach with greater efficacy.

## Figures and Tables

**Figure 1 antibiotics-09-00674-f001:**
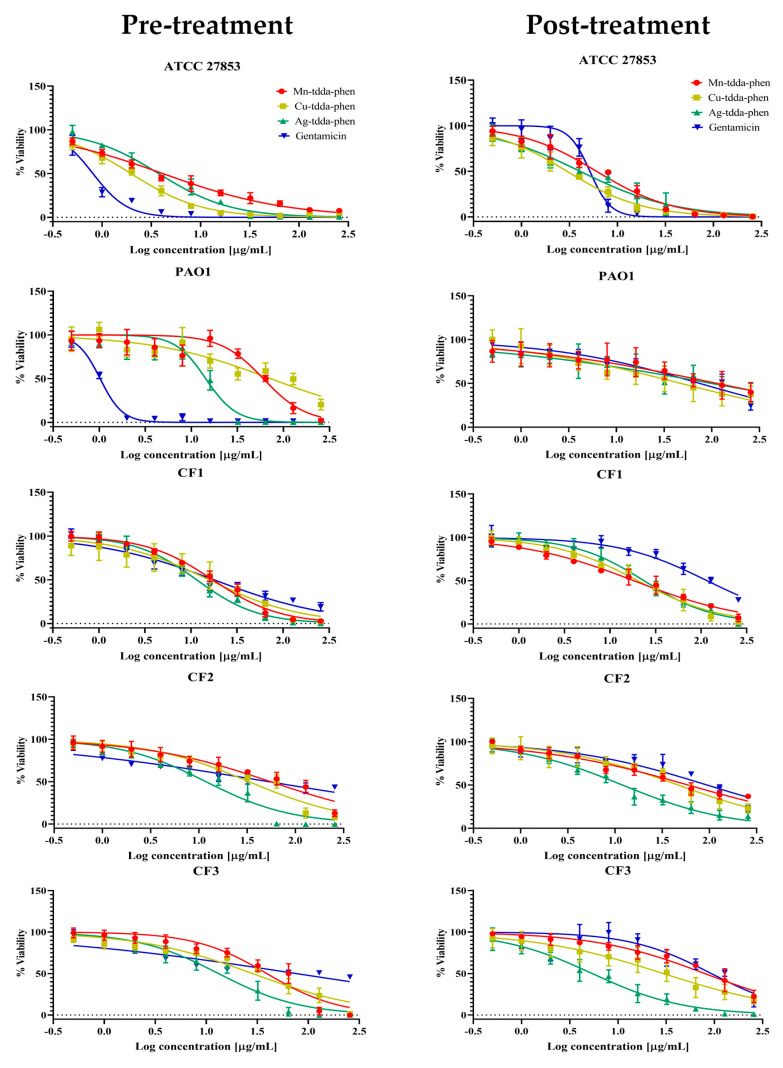
Cell viability—Effect of metal-tdda-phen complexes (Mn-tdda-phen, Cu-tdda-phen and Ag-tdda-phen) and gentamicin on both inhibiting biofilm formation (pre-treatment) and dismantling established biofilms (post-treatment) of laboratory strains (ATCC 27853 and PAO1) and clinical isolates (CF1–CF3).

**Figure 2 antibiotics-09-00674-f002:**
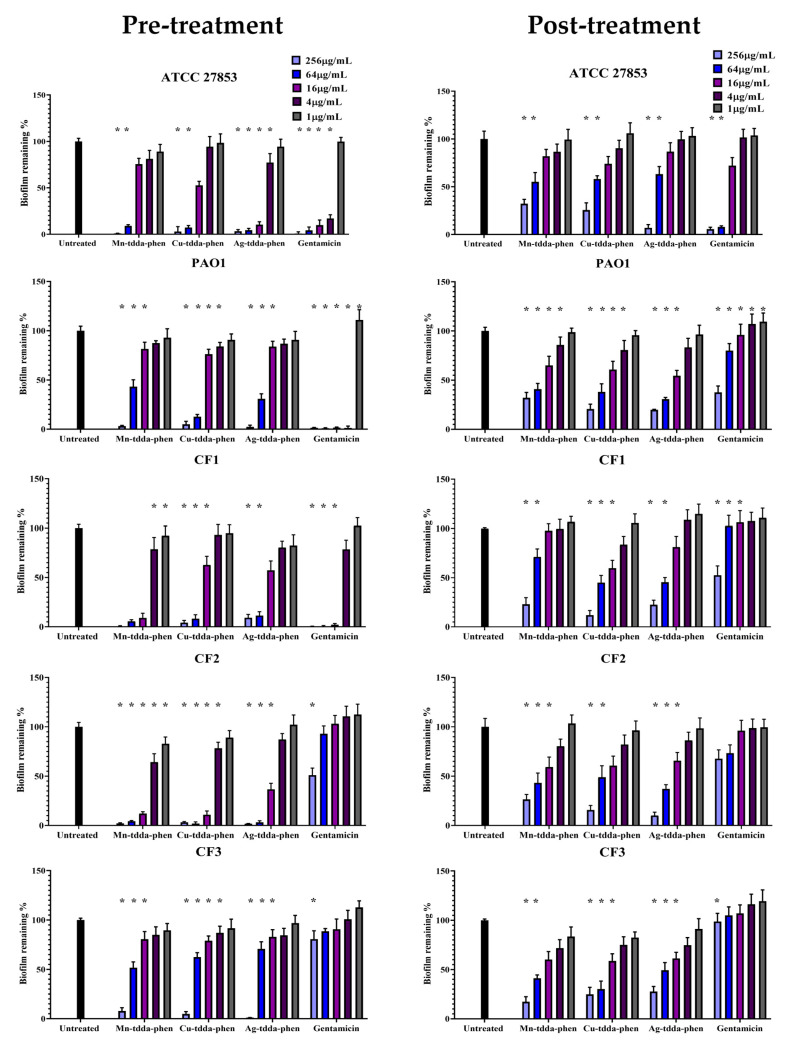
**Biofilm biomass**—Effect of metal-phen complexes (Mn-tdda-phen, Cu-tdda-phen and Ag-tdda-phen) and gentamicin on both inhibiting biofilm formation (pre-treatment) and dismantling established biofilms (post-treatment) of laboratory strains (ATCC 27853 and PAO1) and clinical isolates (CF1–CF3). Asterisks indicate significance (*p* > 0.05) relative to the untreated control.

**Figure 3 antibiotics-09-00674-f003:**
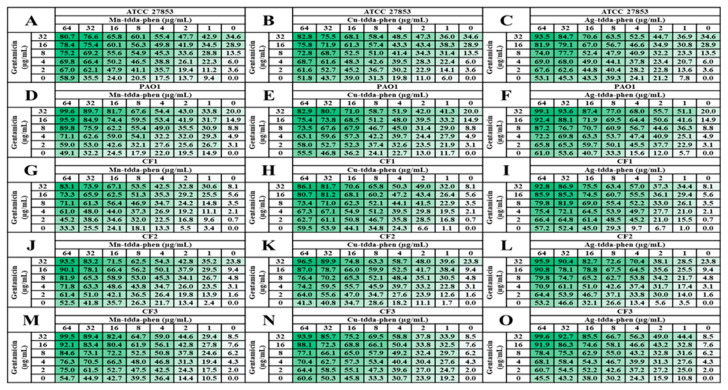
Heat map representation of percentage of biofilm removal of 48 h established biofilms of reference strains, ATCC 27853 and PAO1, and clinical isolates, CF1-CF3, by combinations of metal-tdda-phen complexes with gentamicin. Mn-tdda-phen and gentamicin (**A**,**D**,**G**,**J**,**M**); Cu-tdda-phen and gentamicin (**B**,**E**,**H**,**K**,**N**); Ag-tdda-phen and gentamicin (**C**,**F**,**I**,**L**,**O**). Heat map shows the percentage of biofilm removed with respect to untreated sample.

**Figure 4 antibiotics-09-00674-f004:**
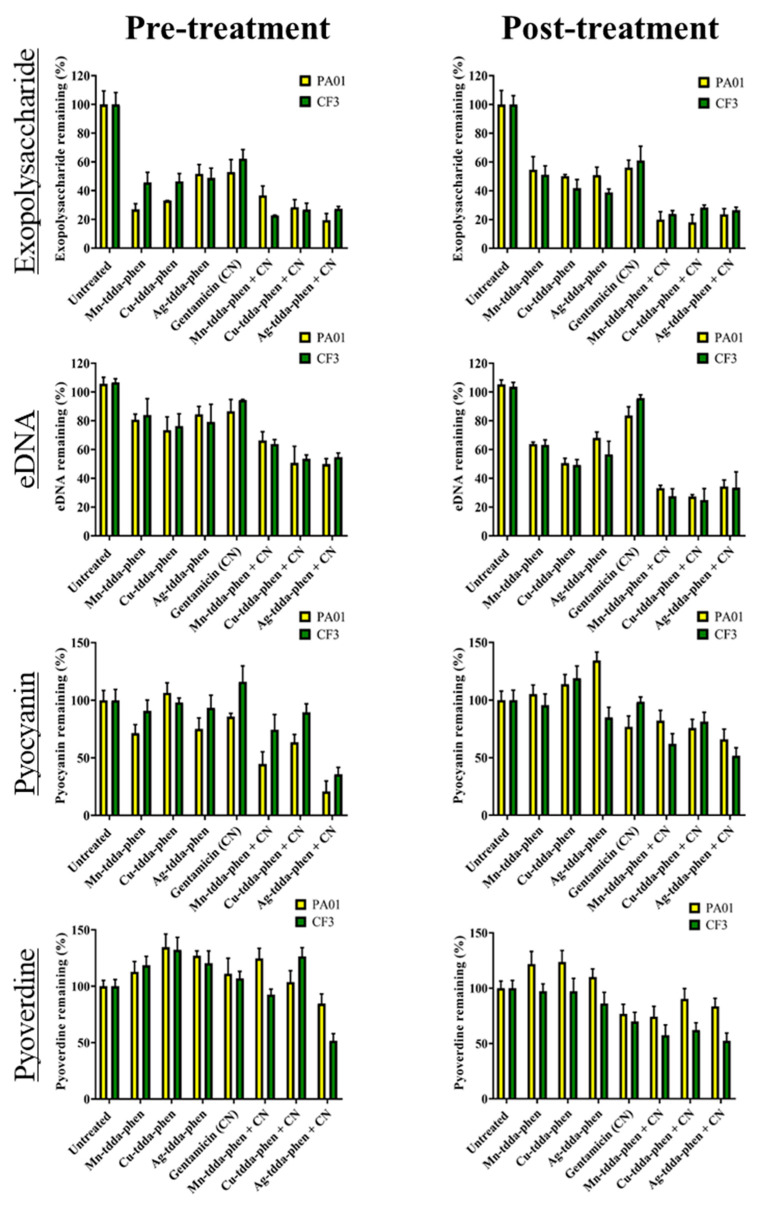
Effects of metal-tdda-phen complexes alone, gentamicin alone, and metal-tdda-phen complexes with gentamicin on both the formation of biofilm (pre-treatment) and reduction of established biofilm (post-treatment) of PAO1 and CF3: The percentages of exopolysaccharide, eDNA, pyocyanin and pyoverdine remaining after treatment (relative to untreated biofilms) are given.

**Figure 5 antibiotics-09-00674-f005:**
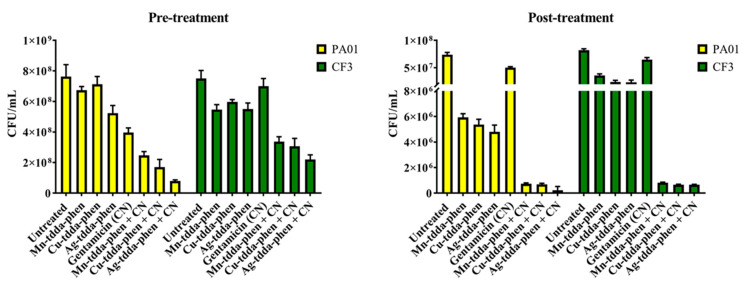
Viable cells were measured after pre- and post- exposure to 0.5 × MBIC and 0.5 × MBEC of metal-tdda-phen complexes alone, gentamicin alone, and metal-tdda-phen complexes with gentamicin. The results are the means ± standard errors of the means from three independent experiments, with viable counts done in triplicate.

**Table 1 antibiotics-09-00674-t001:** Effects of test compounds on planktonic growth of *P. aeruginosa* laboratory strains (ATCC 27853 and PAO1) and clinical isolates (CF1–CF3).

Test Compound	MIC µg/mL and (µM)				
	ATCC 27853	PAO1	CF1	CF2	CF3
Mn-tdda	>256 (737)	>256 (737)	>256 (737)	>256 (737)	>256 (737)
Cu-tdda	>256 (848)	128 (424)	>256 (848)	>256 (848)	256 (848)
Ag-tdda	>256 (542)	>256 (542)	>256 (542)	>256 (542)	>256 (542)
Mn-tdda-phen	16 (21.7)	32 (43.5)	8 (10.9)	8 (10.9)	128 (174)
Cu-tdda-phen	16 (21.5)	32 (43)	8 (10.7)	8 (10.7)	64 (86)
Ag-tdda-phen	8 (6.6)	32 (26.6)	16(13.3)	16 (13.3)	64 (53.2)
Phen	128 (710)	>256 (1420)	128 (710)	128 (710)	>256 (1420)
tddaH_2_	>256 (1152)	>256 (1152)	>256 (1152)	>256 (1152)	>256 (1152)
MnCl_2_	>256 (1294)	>256 (1294)	>256 (1294)	>256 (1294)	>256 (1294)
CuCl_2_	>256 (1502)	>256 (1502)	>256 (1502)	>256 (1502)	>256 (1502)
AgNO_3_	128 (753)	256 (1507)	>256 (1507)	>256 (1507)	256 (1507)
Gentamicin	1 (1.7)	2 (3.5)	8 (13.9)	128 (222)	>256 (445)

**Table 2 antibiotics-09-00674-t002:** Results of interaction studies for metal-tdda-phen/gentamicin combinations against PAO1 (reference strain) and CF3 (clinical isolate) in established biofilms.

Test Complexes	Organism	FIC Index	Interpretation
Mn-tdda-phen + Gentamicin	PAO1	0.141	Synergy
	CF3	0.133	Synergy
Cu-tdda-phen + Gentamicin	PAO1	0.313	Synergy
	CF3	0.156	Synergy
Ag-tdda-phen + Gentamicin	PAO1	0.141	Synergy
	CF3	0.133	Synergy

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
