# Peer review of "The Antibacterial and Anti-Biofilm Activity of Metal Complexes Incorporating 3,6,9-Trioxaundecanedioate and 1,10-Phenanthroline Ligands in Clinical Isolates of Pseudomonas aeruginosa from Irish Cystic Fibrosis Patients"

_antibiotics, 2020, doi:10.3390/antibiotics9100674_

Round 1
Reviewer 1 Report
O'Shaughnessy et al. use chelated tdda-phen complexes to enhance the effects of gentamicin for eliminating biofilms of common laboratory strains and clinical strains of Pseudomonas aerguinosa from CF patients. In light of the concern of growing antibiotic resistance and the lack of viable options to treat these infections, this work is impactful and appropriate for the journal. The experiments are well-controlled and the authors claims are supported by the data. I have a few minor issues that I would like to see addressed before publication.
1) Table 1. Was there any statistical analysis done for these data? It's not readily apparent.
2) Figure 1. All but two of the graphs have a dotted line that runs parallel to X-axis. I'm not sure what this is intended to be?
3) Section 2.3. What is happening to the biofilms? Are the cells actively being killed by the combinatorial method? Are the biofilms dispersing? Would it be possible to take the liquid from these static biofilms and count CFUs? If an experiment cannot be done, I'd like to see the authors expand on what they think is happening because lines 253-256 seem contradictory to me. If the combinations did not have an effect against strains grown in planktonic culture, how can dispersal lead to greater antibiotic efficacy?
4) What are the effects of treatment on pyocyanin in liquid culture?
A few grammatical errors should be fixed and/or clarifications made in the following lines:
4) Lines 96-98 is an incomplete sentence.
5) Lines 195-197 need to be re-worded, it's a confusing sentence.
6) Lines 361-364 should be "demonstrate" not "demonstrates".
7) Lines 426-427 needs some sort of punctuation in it.
Author Response
The authors of Antibiotics-951657 would like to thank reviewer 1 for their valuable time in providing insightful and useful contributions to this manuscript. The inputs received have helped to improve the manuscript and strengthen its impact. We have revised the manuscript and responded to the reviewers in accordance with their remarks.
Please find in attached in our ‘Response to reviewer 1’, the detailed response to the individual comments regarding this manuscript. The response has been written in a comment point counter point format with the reviewers comments in normal font and the authors response in red font. All revisions to the manuscript have been made using track changes.

Reviewer 2 Report
Antibiotics 95165
O’Shaugnessy et al
The authors present three metal chelate complexes as novel anti-microbials against P. aeruginosa, specifically for cystic fibrosis. They are tested on three clinical and two lab strains of Pa. Planktonic growth and biofilm were determined alone and in combination with gentamicin. The three metal complexes were more effective than gentamicin. Combinations of the metal complex and gentamicin were more effective in most cases.
Major comments
- I appreciate the experiment being done with 3 clinical strains. You may wanna explain what the two lab strains are for. I can see that they respond differently to gentamycin and your metal complexes. But why is this difference important if you want to use your anti-microbial in a clinical context. Would it have been sufficient to just study the three clinical strains?
- The manuscript is generally well written. But the authors seem to get lost in numbers and data at times and it is difficult to see the forest for the trees. Minor comments 4 and 5 give you some ideas on summarizing outcomes before you get into the admittedly large amount of data.
Minor comments
- Line 42, 43. I would probably mention somewhere in the first or second sentence that Pa is a human pathogen. It becomes obvious quickly, but right at the beginning it could also be animals.
- I realize there are a lot of data, but the figures are all confusing to read and comprehend. And so is the text. Is there a way to have a short summary at the beginning of each text chapter, so the reader knows what to expect. Mentioning the effect in the subtitle might help, instead of just saying you measured the effect.
- Line 337, Discussion: usually, a discussion starts with a paragraph summarizing the major findings. This first paragraph reads like more Introduction. It takes too long to find out what the outcome of the study is.
Author Response
Response to Reviewer 2
The authors of Antibiotics-951657 would like to thank reviewer 2 for their valuable time in providing insightful and useful contributions to this manuscript. The inputs received have helped to improve the manuscript and strengthen its impact. We have revised the manuscript and responded to the reviewers in accordance with their remarks.
Please find in below in our ‘Response to reviewer 2’, the detailed response to the individual comments regarding this manuscript. The response has been written in a comment point counter point format with the reviewers comments in normal font and the authors response in red font. All revisions to the manuscript have been made using track changes.
